# Is the Perceived Fruit Accessibility Related to Fruit Intakes and Prevalence of Overweight in Disadvantaged Youth: A Cross-Sectional Study

**DOI:** 10.3390/nu12113324

**Published:** 2020-10-29

**Authors:** Narae Yang, Kirang Kim

**Affiliations:** Department of Food Science and Nutrition, College of Natural Sciences, Dankook University, Dandae-ro, Dongnam-gu, Cheonan-si, Chungnam 31116, Korea; skfo2581@daum.net

**Keywords:** food environment, fruit, children, adolescents, obesity, overweight, home, school

## Abstract

Background: Few investigations have studied the relationship between home and school food environments, fruit intakes, and prevalence of overweight in children and adolescents from disadvantaged backgrounds. This study aimed to determine whether food environments for fruit intake at household and school levels affect fruit intakes and risk of overweight among children and adolescents with low household income. Methods: Students (*n* = 3148) in Seoul, Korea completed questionnaires pertaining to select aspects of their food environments, frequency of fruit intakes, and weight status. Chi-square tests and logistic regressions evaluated associations between the aforementioned variables. Results: Participants consumed fruit an average of 0.77 times per day, though its frequency increased when fruit accessibility was perceived positively. The percentage of overweight participants was 23.5% for boys and 22.8% for girls. Generally, fruit intake frequency was linked to a lower prevalence of overweight. Regular provision of fruit in school lunches was associated with a reduced risk of overweight among elementary school girls (odds ratio (OR): 0.52, 95% confidence interval (CI): 0.30–0.92), and having someone at home to prepare fruit was associated with a reduced risk of overweight in elementary school boys (OR: 0.64, 95% CI: 0.43–0.94) and girls (OR: 0.63, 95% CI: 0.43–0.93). Conclusions: The frequency of fruit intake was low among disadvantaged youth. Increasing access to fruit in their food environments appears to enhance consumption and lower the risk of overweight, especially for elementary school girls.

## 1. Introduction

The epidemic of pediatric overweight and obesity has expanded steadily over the past few decades, currently reaching more than 370 million children worldwide [1]. Consequences of early excess adiposity include a heightened risk of psychological problems such as depression, low self-esteem, and disordered eating in the immediate term, as well as cardiovascular disease and various cancers upon adulthood [2,3,4,5]. Adult obesity, implemented in child obesity, makes it difficult to lose weight [6]. Therefore, it is important to prevent obesity in childhood [6]. The prevalence of obesity among children and adolescents has increased in Asia [7]. Although the rising trends of body mass index (BMI) have flattened in high-income countries, the prevalence of obesity is still high in worldwide [7].

Obesity has been explained with reduced intakes of healthy foods and unhealthy eating behaviors [8,9,10,11]. Especially fruit, which are rich in water and fiber, to enhance satiety, and low in energy density [12], have been known to prevent obesity, coronary heart disease, stroke, cardiovascular disease, total cancer, and all-cause mortality [13,14,15,16]. Obesity prevention interventions that take into account healthy food intake, such as fruit, are needed. For effective obesity intervention, it is important to identify the relationship between eating healthy foods and obesity and to find the main determinants that affect obesity. Previous intervention methods for eating healthy foods to prevent obesity have focused on individual behavior changes. However, individual level interventions fundamentally could not change the obesogenic environment, which have been reported to have little effect on long-term improvement [17,18]. In recent years, the ecological model has been applied to promote health in the public health sector. The multilevel interventions that take an account for both individual and environmental factors have been paid attention for effective behavior change. Among environmental factors, food environments include political, economic, social, physical, and natural environment factors which affect accessibility, availability, and affordability of foods [19,20]. The food environments have been known as an important factor related to food choice and dietary intakes [19,20]. For children and adolescents, the household is the first physical and social environment to learn about food intake [21], and the school is responsible for their lunches and another setting to make them eat healthy food [21]. Thus, for effective intervention, factors at household and school levels that affect the availability and accessibility of foods as well as at the individual level should be included [22].

It has been reported that the food environment has a greater effect on health and nutritional status in the vulnerable group than in the general group [23], but most of the studies have been conducted on the general group [24], and there is a dearth of research exploring how food environments at home and school affect fruit intakes and risk of overweight among disadvantaged children and adolescents. Therefore, research is needed to determine how the food environments of vulnerable children is related to healthy food intake and obesity. For this study, we surveyed low socioeconomic status youth to determine how select aspects of their food environments associate with consumption of fruit and weight status. Therefore, the objective of this study was to investigate whether food environments at household and school levels affect fruit intakes and risk of overweight among disadvantaged children and adolescents.

## 2. Methods

### 2.1. Study Population

Participants were recruited in 2015 from the Community Childcare Center, located in Seoul, Korea. They provide welfare services (e.g., protection, education, and meals) after school to youth from disadvantaged backgrounds (e.g., beneficiaries of national basic livelihood or single-parent families, etc.) [25]. During this time, the Community Childcare Center participated in the Healthy Fruit Basket Program, which aims to prevent the development of chronic diseases by providing access to fresh fruits [25,26]. In total, 4154 students (mean age; elementary school: 10.4 y, middle and high school: 14.3 y) were recruited; however, only 3148 were included in the final analyses because they supplied information for all the variables of interest and were of an eligible weight status (i.e., normal and overweight) [27,28]. The Institutional Review Board approved this study, and all participants gave written informed consent and assent (DKU 2015-10-016).

### 2.2. Weight Status

Height and weight were measured by a trained measurer at the public health centers or the Community Childcare Centers using a nationally certified weight and height scale. BMI was calculated as weight (kg) divided by height squared (m^2^). According to the 2007 Korean National Growth Charts for children and adolescents [29], classification of normal weight was defined as having a body weight between the fifth and eighty-fourth percentiles, based on one’s gender and age; classification of overweight was defined as having a body weight in the eighty-fifth percentile or above, or BMI 25 kg/m^2^ or greater, based on one’s gender and age.

### 2.3. Fruit Intakes

Fruit frequency questionnaires examined fruit intakes from the month preceding study enrollment. Scoring ranged from 0 (never) to 9 (3+ times per day). All responses were converted into times per day and collapsed into two categories (<0.5 times per day vs. ≥0.5 times per day) to determine the association with overweight.

### 2.4. Food Environments

Participants were asked five questions about their food environments, adapted from those employed in a previous investigation [30]. Here, three dimensions of the physical environment were evaluated: (1) Availability of fruit at home, (2) accessibility of fruit at home (i.e., is there someone who prepares fruit for children to eat?), and (3) accessibility of fruit at school (i.e., does the school provide fruit twice a week?). For the social environment, questions inquired about the habit of frequently eating fruit among parents and friends. Responses for the questions ranged from 1 (strongly disagree) to 5 (strongly agree), and all were collapsed into categories of disagree (strongly disagree, disagree), neutral, and agree (agree, strongly agree).

### 2.5. Statistical Analysis

Chi-square tests assessed the distribution of participants’ general characteristics, the food environment, and weight status according to the frequency of fruit intakes and aspects of the food environment. T-tests evaluated differences in the frequency of fruit intakes by sex and grade level. An analysis of variance and Scheffe’s post-hoc test determined differences between the food environments. Logistic regressions analyzed the effects of fruit intakes and food environments on weight status, showing odds ratios (ORs), 95% confidence intervals (95% CI). All analyses were performed by SPSS Statistics (v. 23.0); *p* ≤ 0.05 was considered statistically significant.

## 3. Results

General characteristics of the participants are shown in Table 1. Overall, 49.0% were boys and 71.5% were in elementary school. The percentage of overweight for boys and girls was 23.5% and 22.8%, respectively. The percentage of overweight for elementary school, and middle and high school students was 24.3% and 20.4%, respectively. There was a significant difference of weight status between grades, and the percentage of overweight was high in elementary school students.

Table 2 details aspects of the participants’ food environments. Overall, 59.5% to 63.2% of participants responded positively (i.e., affirmed the availability of fruit at home, a regular provision of fruit in school lunches, having someone at home to prepare fruit for them, and a family habit of frequently eating fruit). Between the sexes, girls perceived their food environments more positively than boys (*p* < 0.001). Between the grade levels, elementary school students perceived their food environments more positively than middle and high school students (*p* < 0.001).

Table 3 shows the relationship between aspects of participants’ food environments and frequency of fruit intakes. Regardless of sex or grade level, fruit intakes differed according to how one’s food environment was perceived. Here, the group who perceived their food environments positively was found to consume fruit 0.87 to 0.95 times per day, whereas the group who perceived their food environments negatively was found to consume fruit 0.42 to 0.67 times per day (*p* < 0.001). Interestingly, there were no differences in fruit intakes between sexes when they shared the same perception about their food environments. However, when both grade levels responded positively about their food environments, consumption of fruit was found to be higher among elementary school students. For groups displaying a negative or neutral perception, frequency of fruit intakes was comparable.

Table 4 shows the relationship between aspects of participants’ food environments, frequency of fruit intakes, and weight status. The fruit intake frequency was not related to weight status in total subjects but as classified by grade; the lower intake frequency was shown in overweight groups of elementary school students. In terms of fruit environments and overweight, regular provision of fruit in school lunches and having a person at home to prepare fruit were negatively correlated to an overweight status for all participants. A low frequency of fruit intakes was associated with an overweight status among elementary school students. The proportion of overweight students was lower among those who perceived their food environment at school positively compared to those who did not (*p* = 0.044), especially for girls (*p* = 0.047) and elementary school students (*p* = 0.005). In addition, the proportion of overweight students was lower among those who had someone at home to prepare fruit compared to those who did not (*p* < 0.001), notably again for girls (*p* = 0.007) and elementary school students (*p* = 0.001).

Table 5 shows the relationship between aspects of participants’ food environments and odds of being overweight, unadjusted and adjusted for fruit intake frequency. For boys, those who had someone at home to prepare fruit were unlikely to be overweight (OR = 0.64, 95% CI = 0.43–0.94, *p-trend* = 0.023), and these results remained significant after adjusting for frequency of fruit intakes (OR = 0.63, 95% CI = 0.420.93, *p-trend* = 0.019). For elementary school girls, regular provision of fruit in school lunches (OR = 0.52, 95% CI = 0.30–0.92, *p-trend* = 0.005) and having someone at home to prepare fruit for them (OR = 0.63, 95% CI = 0.43–0.93, *p-trend* = 0.004) were negatively related to an overweight status, even after adjusting for frequency of fruit intakes (OR = 0.54, 95% CI = 0.31–0.95, *p-trend* = 0.01 for regular provision of fruit in school lunches; OR = 0.65, 95% CI = 0.44–0.96, *p-trend* = 0.009 for having someone at home to prepare fruit). Among middle and high school girls, having someone at home to prepare fruit was significantly associated with a lower risk of being overweight after adjusting for frequency of fruit intakes (OR = 0.51, 95% CI = 0.27–0.96, *p-trend* = 0.06).

## 4. Discussion

Because of the prominent influence that food environments exert on an individual’s tendency toward obesity, we explored the relationship between aspects of the home and school food environments, fruit intakes, and overweight status among disadvantaged children and adolescents. We found that the frequency of fruit intakes increased when participants perceived their food environments positively, and this was associated with a reduced prevalence of overweight. In particular, regular provision of fruit in school lunches and having someone at home to prepare fruit was associated with a healthier body weight among elementary school students and girls.

Fruit was consumed an average of 0.77 times per day by our participants, which was below the recommended level of the Dietary Reference Intake for Koreans of at least twice per day [31]. According to the Korea National Health and Nutrition Examination Survey [32], low-income families consume less fruit than medium-income families (100.1 vs. 135.2 g/day). In this study, frequency of fruit intake varied according to a participant’s perception of their food environment, with higher rates of consumption among those who responded positively. These results provide evidence that food environments to increase fruit intakes would play an important role in fruit consumption.

Previously, review papers and meta-analyses have shown that increasing the availability and accessibility of fruits and vegetables for children at school is effective at preventing obesity [33,34]. In this study, regular provision of fruit in school lunches was positively related to fruit intakes among elementary school students, but not middle and high school students. These findings do not align with those from a related investigation in the United States, wherein high school students who received fruits and vegetables from The Fresh Fruit and Vegetable Program were likely to eat fruit more often than those who did not (59.1% vs. 40.9%) [35]. Given that middle and high school students appear to have more established eating habits, it may be necessary to provide fruit at a higher frequency in order to enhance intakes [35,36]. The school lunch service in Korea could provide fruits as a dessert within the school budget, but the current frequency of provision of fruit at school lunch would not be enough to meet the consumption of fruit for the disadvantaged middle and high school students in Korea. Therefore, additional funding for more frequent provision of fruit at school lunch should be needed to increase their fruit consumption.

Regarding the food environment at home, having someone to prepare fruit for students was found to relate favorably to fruit intakes and weight status. This result was consistent with those from other studies reporting that social support for healthy food intakes aides in the prevention of obesity among vulnerable children [37,38]. A few reasons might explain this. First, caregivers give children fruit in a form that allows them to eat it easily. Several studies have shown that providing fruit in a ready-to-eat form or making it visible to children promotes fruit intakes [39,40]. Because the process of washing, cutting, and peeling fruit has been described as an impediment to consumption, providing or storing it in an accessible form may help children eat fruit more frequently [40,41]. Indeed, analyses from the Healthy Habits randomized trial [42] revealed that the frequency of fruit provision from a parent positively impacted their children’s fruit intakes after 12 months. Second, caregivers could promote fruit intakes simply by encouraging their children [43]. Considering that participants in our study hailed from disadvantaged backgrounds, it is unlikely that their caregivers purchase much fruit for them or monitor their intakes at home. Thus, incorporating the Healthy Fruit Basket Program into all Community Childcare Centers and related institutions nationwide may help these individuals increase fruit intakes and attenuate risk of overweight or obesity [25,26].

It is well established that availability of food in the home is a key determinant of consumption among children and adolescents [44,45,46], and our analyses confirmed this phenomenon for fruit, in particular. Additionally, we observed that a family’s habit of eating fruit frequently was connected to high fruit intakes across all participants, similar to findings from other publications [39,42,43].

Interestingly, this investigation detected no difference in fruit intakes between sexes sharing the same perception about their food environments. However, with respect to the grade levels, elementary school students were found to consume fruit more frequently than middle or high school students when their food environment was perceived positively. Childhood is a period of social modeling, as individuals learn how to interact with their environments and behave appropriately [11,47,48]. Hence, the benefits of food environments on fruit intakes may apply more strongly to children than adolescents. For another explanation, a recent study of adults with low socioeconomic position found that self-efficacy on fruit and vegetable consumption was more strongly associated with fruit and vegetable consumption than perceived food environments, which implies the importance of capacity building to partially overcome the poor food environment [49]. As several studies have shown the positive effect of nutritional education on fruit and vegetable intakes [50,51,52], nutritional education intervention should be included for adolescents with negative perception of food environment who especially have low self-efficacy to increase their fruit and vegetable intakes.

When we examined the relationship between fruit intakes and weight status, a high frequency (>0.5 times per day) was tied to a lower prevalence of overweight in girls. According to systematic reviews, consumption of fruit as a means to prevent pediatric obesity did not always produce consistent results in terms of gender or amount (38,39). For example, one investigation noted that eating fruit more than twice a day was only protective against obesity in boys [53], and another showed that excessive fruit intakes actually engendered obesity [54]. Heeding our results and those from earlier studies, consumption of fruit alone may not be adequate to prevent obesity. Specifically, our investigation found that the effects of some aspects of the food environment on the prevalence of obesity remained significant after adjusting for fruit intake frequency, and a related study observed a positive link between parental concern for their children’s diet and fitness practices (*r* = 0.552, *p* < 0.001) [55]. It could suggest that the home environment supporting children to eat fruits could also support other behaviors that can prevent obesity in children, such as encouraging exercise. Further research exploring caretakers’ interest in their children’s health will provide a more holistic understanding of how environments moderate their risk of obesity.

The present study had several limitations and strengths. Concerning limitations, this was a cross-sectional study, so any causal associations between food environments, fruit intakes, and overweight status have yet to be determined. In addition, the questionnaire evaluating fruit availability and accessibility in the home and school food environments was not validated, although it had been employed in a previous study [30]. Moreover, we did not collect information pertaining to other confounders of overweight, which could offer deeper insight into factors underlying the present findings [25,26]. Granted, this was a highly homogenous cohort since all participants were recruited from the Community Childcare Center, so it is unlikely they possessed any remarkable characteristics that would alter our results. Nonetheless, future studies examining a more diverse population are needed to confirm the relationship between the food environment and risk of obesity by adjusting the confounding factor. Despite limitations, these results provide evidence that food environments play an important role in overweight prevention as well as fruit consumption among underprivileged Asian children and adolescents, using a relatively large sample size.

## 5. Conclusions

This study found that the frequency of fruit intakes was generally low among children and adolescents from disadvantaged backgrounds; however, those who perceived their food environments positively consumed more fruit and were less likely to be overweight than those who perceived them negatively. Noteworthily, regular provision of fruit in school lunches and having someone at home to prepare fruit were predictive of a high frequency of fruit intakes and lower prevalence of overweight, and these discoveries were most apparent among elementary school students and girls. On the whole, our findings demonstrate that augmenting access to fruit within any realm of the food environment is associated with increased consumption and healthier body weights for low-income youth. Going forward, the school lunch service in Korea could consider increasing fruit servings since current provisions are not sufficient for enabling disadvantaged middle and high school students to meet the dietary guidelines. In addition, a nation-wide program to increase fruit consumption at the Community Childcare Center where they usually spend time would be essential to improve their health.

## Figures and Tables

**Table 1 nutrients-12-03324-t001:** General characteristics of subjects ^1^.

	All (*n* = 3148)	Normal Weight (*n* = 2419)	Overweight (*n* = 729)	*p* ^2^
Sex				
Boys	1542 (49.0)	1179 (76.5)	363 (23.5)	0.617
Girls	1606 (51.0)	1240 (77.2)	366 (22.8)	
Grade				
Elementary school	2251 (71.5)	1705 (75.7)	546 (24.3)	0.021
Middle and high school	897 (28.5)	714 (79.6)	183 (20.4)	
Sex, grade				
Boys				
Elementary school	1061 (68.8)	797 (75.1)	264 (24.9)	0.065
Middle and high school	481 (31.2)	382 (79.4)	99 (20.6)	
Girls				
Elementary school	1190 (74.1)	908 (76.3)	282 (23.7)	0.142
Middle and high school	416 (25.9)	332 (79.8)	84 (20.2)	

^1^ Qualitative variables are presented as n (%). ^2^
*p*-values for differences between the weight statuses were obtained by a chi-square test.

**Table 2 nutrients-12-03324-t002:** Select aspects of participants’ food environments ^1^.

	All	Sex		Grade	
		Boys	Girls	Elementary School	Middle and High School
Availability of fruit at home					
Disagree	458 (14.5)	238 (15.4)	220 (13.7)	313 (13.9)	145 (16.2)
Neutral	816 (25.9)	441 (28.6)	375 (23.3)	557 (24.7)	259 (28.9)
Agree	1874 (59.5)	863 (56.0)	1011 (63.0)	1381 (61.4)	493 (55.0)
*p* ^2^		<0.001		0.004	
Regular provision of fruit in school lunches					
Disagree	259 (8.2)	155 (10.1)	104 (6.5)	145 (6.4)	114 (12.7)
Neutral	959 (30.5)	504 (32.7)	455 (28.3)	624 (27.7)	335 (37.3)
Agree	1930 (61.3)	883 (57.3)	1047 (65.2)	1482 (65.8)	448 (49.9)
*p* ^2^		<0.001		<0.001	
Having someone at home to prepare fruit					
Disagree	439 (13.9)	226 (14.7)	213 (13.3)	295 (13.1)	144 (16.1)
Neutral	718 (22.8)	393 (25.5)	325 (20.2)	471 (20.9)	247 (27.5)
Agree	1991 (63.2)	923 (59.9)	1068 (66.5)	1485 (66.0)	506 (56.4)
*p* ^2^		<0.001		<0.001	
Family habit of eating fruit frequently					
Disagree	282 (9.0)	143 (9.3)	139 (8.7)	193 (8.6)	89 (9.9)
Neutral	954 (30.3)	518 (33.6)	436 (27.1)	618 (27.5)	336 (37.5)
Agree	1912 (60.7)	881 (57.1)	1031 (64.2)	1440 (64.0)	472 (52.6)
*p* ^2^		<0.001		<0.001	
Friends’ habit of eating fruit frequently					
Disagree	544 (17.3)	299 (19.4)	245 (15.3)	364 (16.2)	180 (20.1)
Neutral	1378 (43.8)	710 (46.0)	668 (41.6)	936 (41.6)	442 (49.3)
Agree	1226 (38.9)	533 (34.6)	693 (43.2)	951 (42.2)	275 (30.7)
*p* ^2^		<0.001		<0.001	

^1^ Values are presented as absolute numbers and percentages. ^2^
*p*-values for differences between sexes or grade levels, obtained by a chi-square test.

**Table 3 nutrients-12-03324-t003:** Relationship between aspects of participants’ food environments and frequency of fruit intakes ^1^.

	All	Sex		Grade	
		Boys	Girls	Elementary School	Middle and High School
All	0.77 ± 0.17	0.74 ± 0.02 ^a^	0.80 ± 0.02 ^b^	0.85 ± 0.02 ^a^	0.59 ± 0.02 ^b^
*p* ^2^		0.036			<0.001
Availability of fruit at home					
Disagree	0.47 ± 0.03 ^a^	0.46 ± 0.04 ^a^	0.49 ± 0.04 ^a^	0.51 ± 0.04 ^a^	0.39 ± 0.03 ^a^
Neutral	0.55 ± 0.02 ^a^	0.54 ± 0.03 ^a^	0.55 ± 0.03 ^a^	0.60 ± 0.03 ^ab^	0.44 ± 0.03 ^a^
Agree	0.95 ± 0.02 ^b^	0.92 ± 0.03 ^b^	0.96 ± 0.03 ^b^	1.03 ± 0.03 ^c^	0.71 ± 0.03 ^b^
*p* ^2^	<0.001	<0.001		<0.001	
Regular provision of fruit in school lunches					
Disagree	0.65 ± 0.05 ^a^	0.62 ± 0.06 ^a^	0.69 ± 0.08 ^ab^	0.64 ± 0.07 ^acd^	0.66 ± 0.06 ^ad^
Neutral	0.62 ± 0.02 ^a^	0.62 ± 0.03 ^a^	0.63 ± 0.03 ^a^	0.70 ± 0.03 ^a^	0.47 ± 0.03 ^cd^
Agree	0.87 ± 0.02 ^b^	0.83 ± 0.03 ^b^	0.89 ± 0.03 ^b^	0.93 ± 0.02 ^b^	0.64 ± 0.03 ^ad^
*p* ^2^	<0.001	<0.001		<0.001	
Having someone at home to prepare fruit					
Disagree	0.60 ± 0.04 ^a^	0.55 ± 0.05 ^a^	0.64 ± 0.05 ^a^	0.67 ± 0.05 ^ae^	0.43 ± 0.03 ^ad^
Neutral	0.53 ± 0.02 ^a^	0.52 ± 0.03 ^a^	0.55 ± 0.04 ^a^	0.58 ± 0.03 ^ac^	0.44 ± 0.03 ^cd^
Agree	0.90 ± 0.02 ^b^	0.88 ± 0.03 ^b^	0.91 ± 0.03 ^b^	0.97 ± 0.02 ^b^	0.70 ± 0.03 ^e^
*p* ^2^	<0.001	<0.001		<0.001	
Family habit of eating fruit frequently					
Disagree	0.42 ± 0.04 ^a^	0.42 ± 0.05 ^a^	0.43 ± 0.05 ^a^	0.46 ± 0.05 ^ab^	0.33 ± 0.03 ^ab^
Neutral	0.54 ± 0.02 ^a^	0.55 ± 0.03 ^a^	0.53 ± 0.03 ^a^	0.59 ± 0.03 ^bc^	0.45 ± 0.03 ^ab^
Agree	0.94 ± 0.02 ^b^	0.91 ± 0.03 ^b^	0.97 ± 0.03 ^b^	1.01 ± 0.02 ^d^	0.72 ± 0.03 ^c^
*p* ^2^	<0.001	<0.001		<0.001	
Friends’ habit of eating fruit frequently					
Disagree	0.67 ± 0.03 ^a^	0.63 ± 0.04 ^a^	0.71 ± 0.05 ^ab^	0.75 ± 0.04 ^a^	0.49 ± 0.03 ^bd^
Neutral	0.67 ± 0.02 ^a^	0.68 ± 0.03 ^a^	0.67 ± 0.03 ^a^	0.73 ± 0.03 ^a^	0.56 ± 0.03 ^bd^
Agree	0.93 ± 0.03 ^b^	0.89 ± 0.04 ^bc^	0.96 ± 0.04 ^c^	1.00 ± 0.03 ^c^	0.69 ± 0.04 ^ad^
*p* ^2^	<0.001	<0.001		<0.001	

^1^ Values are fruit intake per day, presenting as means ± standard errors. ^2^
*p*-values for differences between sexes or grade levels, obtained by ANOVA; letters (^a,b,c,d,e^) indicate significant differences between groups (Scheffe’s post-hoc test, *p* < 0.05).

**Table 4 nutrients-12-03324-t004:** Relationship between aspects of participants’ food environments, frequency of fruit intakes, and weight status ^1^.

	All		Sex				Grade			
			Boys		Girls		Elementary School		Middle and High School	
	Normal Weight	Over Weight	Normal Weight	Over Weight	Normal Weight	Over Weight	Normal Weight	Over Weight	Normal Weight	Over Weight
**Fruits intakes per day**										
<0.5	1194 (49.4)	378 (51.9)	610 (51.7)	191 (52.6)	584 (47.1)	187 (51.1)	771 (45.2)	274 (50.2)	423 (59.2)	104 (56.8)
≥0.5	1225 (50.6)	351 (48.1)	569 (48.3)	172 (47.4)	656 (52.9)	179 (48.9)	934 (54.8)	272 (49.8)	291 (40.8)	79 (43.2)
*p* ^2^	0.238		0.770		0.179		0.043		0.554	
**Food environments**										
Availability of fruit at home										
Disagree	348 (14.4)	110 (15.1)	181 (15.4)	57 (15.7)	167 (13.5)	53 (14.5)	232 (13.6)	81 (14.8)	116 (16.2)	29 (15.8)
Neutral	629 (26.0)	187 (25.7)	335 (28.4)	106 (29.2)	294 (23.7)	81 (22.1)	420 (24.6)	137 (25.1)	209 (29.3)	50 (27.3)
Agree	1442 (59.6)	432 (59.3)	663 (56.2)	200 (55.1)	779 (62.8)	232 (63.4)	1053 (61.8)	328 (60.1)	389 (54.5)	104 (56.8)
*p* ^2^	0.892		0.929		0.768		0.713		0.838	
Regular provision of fruit in school lunches										
Disagree	187 (7.7)	72 (9.9)	115 (9.8)	40 (11.0)	72 (5.8)	32 (8.7)	99 (5.8)	46 (8.4)	88 (12.3)	26 (14.2)
Neutral	723 (29.9)	236 (32.4)	380 (32.2)	124 (34.2)	343 (27.7)	112 (30.6)	454 (26.6)	170 (31.1)	269 (37.7)	66 (36.1)
Agree	1509 (62.4)	421 (57.8)	684 (58.0)	199 (54.8)	825 (66.5)	222 (60.7)	1152 (67.6)	330 (60.4)	357 (50.0)	91 (49.7)
*p* ^2^	0.044		0.535		0.047		0.005		0.775	
Having someone at home to prepare fruit										
Disagree	310 (12.8)	129 (17.7)	160 (13.6)	66 (18.2)	150 (12.1)	63 (17.2)	204 (12.0)	91 (16.7)	106 (14.8)	38 (20.8)
Neutral	539 (22.3)	179 (24.6)	297 (25.2)	96 (26.4)	242 (19.5)	83 (22.7)	342 (20.1)	129 (23.6)	197 (27.6)	50 (27.3)
Agree	1570 (64.9)	421 (57.8)	722 (61.2)	201 (55.4)	848 (68.4)	220 (60.1)	1159 (68.0)	326 (59.7)	411 (57.6)	95 (51.9)
*p* ^2^	<0.001		0.055		0.007		0.001		0.135	
Family habit of eating fruit frequently										
Disagree	211 (8.7)	71 (9.7)	107 (9.1)	36 (9.9)	104 (8.4)	35 (9.6)	141 (8.3)	52 (9.5)	70 (9.8)	19 (10.4)
Neutral	730 (30.2)	224 (30.7)	393 (33.3)	125 (34.4)	337 (27.2)	99 (27.0)	454 (26.6)	164 (30.0)	276 (38.7)	60 (32.8)
Agree	1478 (61.1)	434 (59.5)	679 (57.6)	202 (55.6)	799 (64.4)	232 (63.4)	1110 (65.1)	330 (60.4)	368 (51.5)	104 (56.8)
*p* ^2^	0.629		0.780		0.778		0.141		0.339	
Friends’ habit of eating fruit frequently										
Disagree	421 (17.4)	123 (16.9)	233 (19.8)	66 (18.2)	188 (15.2)	57 (15.6)	275 (16.1)	89 (16.3)	146 (20.4)	34 (18.6)
Neutral	1051 (43.4)	327 (44.9)	541 (45.9)	169 (46.6)	510 (41.1)	158 (43.2)	697 (40.9)	239 (43.8)	354 (49.6)	88 (48.1)
Agree	947 (39.1)	279 (38.3)	405 (34.4)	128 (35.3)	542 (43.7)	151 (41.3)	733 (43.0)	218 (39.9)	214 (30.0)	61 (33.3)
*p* ^2^	0.796		0.798		0.702		0.415		0.652	

^1^ Values are presented as absolute numbers and percentages. ^2^
*p*-values for differences between sexes and grade levels, obtained by a chi-square test.

**Table 5 nutrients-12-03324-t005:** Relationship between aspects of participants’ food environments and odds of being overweight.

	Boys				Girls			
	Elementary School		Middle and High School		Elementary School		Middle and High School	
	OR ^1^ (95% CI)	Adjusted OR ^2^ (A95% CI)	OR (95% CI)	Adjusted OR (A95% CI)	OR (95% CI)	Adjusted OR (A95% CI)	OR (95% CI)	Adjusted OR (A95% CI)
Availability of fruit at home								
Disagree	1.00	1.00	1.00	1.00	1.00	1.00	1.00	1.00
Neutral	0.87 (0.56–1.35)	0.87 (0.56–1.34)	1.41 (0.69–2.90)	1.41 (0.69–2.88)	1 (0.63–1.60)	1.02 (0.64–1.62)	0.61 (0.29–1.29)	0.61 (0.29–1.27)
Agree	0.8 (0.54–1.19)	0.79 (0.53–1.18)	1.44 (0.74–2.80)	1.41 (0.72–2.77)	1 (0.67–1.50)	1.08 (0.72–1.63)	0.77 (0.40–1.48)	0.7 (0.36–1.37)
*p-trend* ^3^	0.278	0.246	0.342	0.392	0.988	0.652	0.701	0.464
Regular provision of fruit in school lunches								
Disagree	1.00	1.00	1.00	1.00	1.00	1.00	1.00	1.00
Neutral	0.87 (0.52–1.47)	0.87 (0.52–1.47)	0.98 (0.49–1.96)	1 (0.50–1.99)	0.72 (0.39–1.30)	0.72 (0.40–1.31)	0.66 (0.30–1.44)	0.71 (0.32–1.55)
Agree	0.71 (0.43–1.16)	0.7 (0.43–1.15)	1.01 (0.52–1.98)	1.01 (0.52–1.98)	0.52 (0.30–0.92)	0.54 (0.31–0.95)	0.7 (0.34–1.46)	0.71 (0.34–1.49)
*p-trend* ^3^	0.081	0.075	0.939	0.963	0.005	0.01	0.527	0.492
Having someone at home to prepare fruit								
Disagree	1.00	1.00	1.00	1.00	1.00	1.00	1.00	1.00
Neutral	0.76 (0.49–1.19)	0.76 (0.49–1.19)	0.85 (0.44–1.64)	0.85 (0.45–1.64)	0.95 (0.60–1.50)	0.93 (0.59–1.47)	0.56 (0.27–1.16)	0.55 (0.27–1.14)
Agree	0.64 (0.43–0.94)	0.63 (0.42–0.93)	0.72 (0.40–1.32)	0.7 (0.38–1.29)	0.63 (0.43–0.93)	0.65 (0.44–0.96)	0.57 (0.30–1.05)	0.51 (0.27–0.96)
*p-trend* ^3^	0.023	0.019	0.275	0.223	0.004	0.009	0.122	0.06
Family habit of eating fruit frequently								
Disagree	1.00	1.00	1.00	1.00	1.00	1.00	1.00	1.00
Neutral	1.16 (0.69–1.93)	1.15 (0.68–1.92)	0.63 (0.29–1.36)	0.63 (0.29–1.36)	0.81 (0.48–1.37)	0.82 (0.49–1.38)	1.08 (0.45–2.58)	1.03 (0.43–2.46)
Agree	0.83 (0.51–1.36)	0.81 (0.49–1.33)	1.01 (0.48–2.12)	1.00 (0.47–2.13)	0.78 (0.49–1.25)	0.84 (0.52–1.36)	1.09 (0.47–2.50)	0.96 (0.41–2.25)
*p-trend* ^3^	0.097	0.081	0.301	0.346	0.357	0.649	0.879	0.835
Friends’ habit of eating fruit frequently								
Disagree	1.00	1.00	1.00	1.00	1.00	1.00	1.00	1.00
Neutral	1.06 (0.72–1.56)	1.06 (0.72–1.56)	1.26 (0.69–2.31)	1.25 (0.69–2.29)	1.06 (0.70–1.59)	1.04 (0.69–1.58)	0.86 (0.45–1.65)	0.85 (0.44–1.63)
Agree	1.02 (0.69–1.52)	1.02 (0.69–1.51)	1.31 (0.67–2.57)	1.29 (0.65–2.54)	0.84 (0.56–1.27)	0.86 (0.57–1.31)	1.13 (0.59–2.17)	1.07 (0.55–2.08)
*p-trend* ^3^	0.962	0.977	0.459	0.496	0.204	0.302	0.586	0.699

^1^ Odds ratios (ORs) were derived from logistic regression models. ^2^ Adjusted ORs adjusted for fruit intake frequency and were derived from logistic regression models. ^3^
*p*-values for trends.

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
