# Peer review of "Is the Perceived Fruit Accessibility Related to Fruit Intakes and Prevalence of Overweight in Disadvantaged Youth: A Cross-Sectional Study"

_nutrients, 2020, doi:10.3390/nu12113324_

Round 1

Reviewer 1 Report

A few studies have investigated the relationship between eating habits at home and at school with regards to fruit intake and the risk of becoming overweight in children and adolescents from disadvantaged backgrounds, especially those of Eastern Countries.

It is interesting to note how the study by Yang and Kim aims to determine whether eating habits at home or at school influence fruit intake and the risk of becoming overweight among children and adolescents in disadvantaged contexts in Korea.

The study is based on a large number of participants, with a total of 4,154 students recruited and 3,148 included in the analyses.

As discussed by the authors, the cross-sectional design of the study does not allow the identification of the causal association between eating habits, fruit intake and becoming overweight. Furthermore, the questionnaire used to assess the availability and accessibility of fruit at home and at school, has not been validated, although used in a previous study.

In addition, no information was collected regarding other confounding factors linked to becoming overweight (e.g. physical activity, diet composition), which could help in giving a more comprehensive interpretation of the results obtained.

Despite the limitations, this study, with the use of a large sample, provides evidence that food environment and the easy access to fruit plays an important role in fruit consumption and in the prevention of becoming overweight among disadvantaged Asian children and adolescents.

It is recommended that recent papers of interest should be discussed in the appropriate section.

M.G. Wagner. Nutrition education effective in increasing fruit and vegetable consumption among overweight and obese adults (2016). https://pubmed.ncbi.nlm.nih.gov/26850310/

M.C de Menezes. Fruit and vegetable intake: Influence of perceived food environment and self-efficacy (2018). Link https://pubmed.ncbi.nlm.nih.gov/29753052/

Honrath K. Does Nutrition Education with Fruit and Vegetable Supplementation Increase Fruit and Vegetable Intake and Improve Anthropometrics of Overweight or Obese People of Varying Socioeconomic Status? (2018) Link https://pubmed.ncbi.nlm.nih.gov/29192798/

R.L. Clark Educational intervention improves fruit and vegetable intake in young adults with metabolic syndrome components (2019). Link https://pubmed.ncbi.nlm.nih.gov/30803510/

A. Ridberg. Effect of a Fruit and Vegetable Prescription Program on Children's Fruit and Vegetable Consumption. Link https://pubmed.ncbi.nlm.nih.gov/31198165/

Author Response

We added the reference as you suggested (Line # 206) and we added the discussion on the nutritional education to increase self-efficacy for the F&V intake. Thank you for your suggestion (Line # 234-240).

Reviewer 2 Report

The paper aims to investigate whether food environments at household and school-level affect fruit intakes and the risk of overweight among disadvantaged children and adolescents. A sample of 4154 students of elementary school, middle and high school were included in the study. Participants consumed fruit an average of 0.77 times per day, though its frequency increased when fruit accessibility was perceived positively. The percentage of overweight was 23.5% for boys and 22.8% for girls. Generally, fruit intake frequency was linked to a lower prevalence of overweight. Regular availability of fruit in school lunches was associated with a reduced risk of overweight among elementary school girls; therefore, having someone at home to prepare fruit was associated with a reduced risk of overweight in elementary school boys and girls. In conclusion, the authors suggest that increasing the availability of fruits in their foods environments could promote a better lifestyle.

The reading of the paper suggests some comments:

  • This paper has been found of some interest in its field, considering the important health implications of diet and lifestyle among children and adolescents. The research question posed by the authors is easily identifiable and understood. The study design was in the complex carefully carried out.
  • Consider in the Introduction Section a recent work: “Mediterranean diet, nutrition transition, and cardiovascular risk factor in children and adolescents”, Roberta Ricotti, Marina Caputo, Flavia Prodam, published in “The Mediterranean Diet: An Evidence-Based Approach” edited by Second Edition (2020), Victor R. Preedy, Ronald Ross Watson.
  • Methods: 1) Specify instruments for height and weight measurements; also describe how to calculate body mass index. 2) Was there a trained medical staff for anthropometrical measurements? 3) The overall sample was classified in the overweight and obese group, according to body mass index, even update those with the new cut-offs proposed recently by The International Obesity Task Force (IOTF, Pediatric Obesity, 2012). 4) Attach Questionnaires administered to subjects for the reproducibility of the study; are Questionnaires used validated? 5) Specify how was defined the socio-economic level of families included in the study.

Author Response

Point 1: Consider in the Introduction Section a recent work: “Mediterranean diet, nutrition transition, and cardiovascular risk factor in children and adolescents”, Roberta Ricotti, Marina Caputo, Flavia Prodam, published in “The Mediterranean Diet: An Evidence-Based Approach” edited by Second Edition (2020), Victor R. Preedy, Ronald Ross Watson.

Response: Thank you for suggesting recent work to consider. We added the reference as you suggested (Line # 38).

Point 2: Specify instruments for height and weight measurements; also describe how to calculate body mass index. Was there a trained medical staff for anthropometrical measurements?

Response: Height and weight were measured in public health centers or community childcare centers. Although the measuring device was different, we assume that there would be no significant measurement error because nationally certified scales were used to measure height and weight.

We added one paragraph in the methods section (Line # 77-79).

Point 3: The overall sample was classified in the overweight and obese group, according to body mass index, even update those with the new cut-offs proposed recently by The International Obesity Task Force (IOTF, Pediatric Obesity, 2012).

Response: The body mass index cut-off of the International Obesity Task Force was calculated from various countries, and the criteria we have considered are the criteria for Korean children calculated using the LMS method for Koreans. This criterion would be considered to be more valid in screening overweight and obesity in Korean children population.

Point 4: Attach Questionnaires administered to subjects for the reproducibility of the study; are Questionnaires used validated?

Response: In our discussion, we mentioned that the limitation of our study is that the questionnaire evaluating the home and school food environments was not validated, and the questionnaire is attached as a supplementary appendix (Line # 418).

Point 5: Specify how was defined the socio-economic level of families included in the study.

Response: Mostly, households using the Community Childcare Center were beneficiaries of national basic livelihood or single-parent families. To specify the socio-economic level, this content was added to the methods section (Line # 69-70).
